# The Possible Synergistic Pharmacological Effect of an Oral Berberine (BBR) and Curcumin (CUR) Complementary Therapy Alleviates Symptoms of Irritable Bowel Syndrome (IBS): Results from a Real-Life, Routine Clinical Practice Settings-Based Study

**DOI:** 10.3390/nu16081204

**Published:** 2024-04-18

**Authors:** Ursula Wade, Domingo A. Pascual-Figal, Fazale Rabbani, Marie Ernst, Adelin Albert, Isabelle Janssens, Yvan Dierckxsens, Somia Iqtadar, Nisar A. Khokhar, Ayesha Kanwal, Amjad Khan

**Affiliations:** 1Department of Basic and Clinical Neuroscience, Kings College London, London SE5 9RT, UK; ursulawade38@gmail.com; 2Hospital Universitario Virgen de la Arrixaca, IMIB-Arrixaca, Universidad de Murcia, 30120 Murcia, Spain; dpascual@um.es; 3Lady Reading Hospital, Peshawar 25000, Pakistan; fazlerabbani@lrh.edu.pk (F.R.); ayesha.kanwal@lrh.edu.pk (A.K.); 4Biostatistics and Research Methods Center (B-STAT), CHU of Liège and University of Liège, 4000 Liège, Belgiumaalbert@uliege.be (A.A.); 5Laboratoire Tilman, 5377 Baillonville, Belgium; ijanssens@tilman.be (I.J.); yd@tilman.be (Y.D.); 6Department of Medicine, King Edward Medical University, Lahore 54000, Pakistan; somia.iqtadar@gmail.com; 7Department of Medicine, Bilawal Medical College, Liaquat University of Medical and Health Sciences, Jamshoro 76090, Pakistan; dr.nisarkhokhar@gmail.com; 8Department of Biochemistry, Liaquat University of Medical and Health Sciences, Jamshoro 76090, Pakistan; 9Department of Oncology, University of Oxford, Oxford OX3 7DQ, UK

**Keywords:** irritable bowel syndrome (IBS), berberine, curcumin, Enterofytol^®^ PLUS

## Abstract

Irritable bowel syndrome (IBS) is a prevalent chronic functional gastrointestinal disorder, characterised by recurrent abdominal discomfort and altered bowel movements. IBS cause a significantly negative impact on quality of life (QoL). Growing pharmacological evidence suggests that berberine (BBR) and curcumin (CUR) may mitigate IBS symptoms through multiple complementary synergistic mechanisms, resulting in the attenuation of intestinal inflammation and regulation of bowel motility and gut functions. In the present observational study conducted under real-life routine clinical practice settings, 146 patients diagnosed with IBS were enrolled by general practitioner clinics and pharmacies in Belgium. For the first time, this study assessed the potential synergistic pharmacological effect of a combined oral BBR/CUR supplement (Enterofytol^®^ PLUS, containing 200 mg BBR and 49 mg CUR) (two tablets daily for 2 months), serving as complementary therapy in the management of IBS. Following the 2-month supplementation, significant improvements were observed in the patients’ IBS severity index (IBSSI) (47.5%) and all the primary IBS symptoms, such as abdominal discomfort (47.2%), distension (48.0%), intestinal transit (46.8%), and QoL (48.1%) (all *p* < 0.0001). The improvement in the patients’ IBSSI was independent of age, sex, and IBS sub-types. The patients’ weekly maximum stool passage frequency decreased significantly (*p* < 0.0001), and the stool status normalized (*p* < 0.0001). The patients’ need for concomitant conventional IBS treatment decreased notably: antispasmodics by 64.0% and antidiarrhoeals by 64.6%. Minor adverse effects were reported by a small proportion (7.1%) of patients, mostly gastrointestinal. The majority (93.1%) experienced symptom improvement or resolution, with a high satisfaction rate (82.6%) and willingness to continue the supplementation (79.0%). These findings support the potential synergistic pharmacological role of BBR and CUR in IBS, and their co-supplementation may alleviate IBS symptoms and improve QoL.

## 1. Introduction

Irritable bowel syndrome (IBS) is a chronic functional gastrointestinal (GI) disorder characterised by recurrent abdominal discomfort associated with altered bowel movements [1,2,3]. The IBS abdominal discomfort may present in different forms, such as sharp pain, distention, bloating, cramping, fullness, or even burning [1,2,3]. These abdominal conditions may be triggered following a meal, eating specific foods, emotional stress, constipation, or diarrhoea [1,2,3]. Other GI symptoms associated with IBS may include mucus in the stool, faecal urgency, and persistent rectal tenesmus (feeling of incomplete bowel emptiness) [1,2,3]. People with IBS may also experience other symptoms not directly related to the bowel, such as migraine headaches, sleep disturbances, and feelings of anxiety or depression [1,2,3]. IBS is diagnosed in accordance with the Rome IV criteria, defined by recurring abdominal discomfort (pain) alongside two or more of the following criteria: changes in defecation patterns, alterations in stool frequency, or variations in stool appearance [1,2,3]. Symptoms must manifest with a minimum frequency of at least once per week over the preceding 3 months, with a duration of no less than 6 months [1,2,3]. IBS has four sub-types according to the predominant bowel habits that individuals experience, i.e., IBS with constipation (IBS-C), IBS with diarrhoea (IBS-D), mixed IBS (with alternating diarrhoea and constipation), and unspecified IBS, if it does not correspond to any other type [1,2,3,4]. Women are up to two times more likely to develop IBS than men, and the condition predominantly affects individuals under the age of 65. IBS significantly impairs individuals’ quality of life (QoL), causing in reduced work productivity, social burdens, and heightened reliance on healthcare resources [5].

Despite the high global prevalence of 11.2% [6], the pathophysiology of IBS remains insufficiently elucidated and seems to be multifactorial. However, increased intestinal mucosal permeability [7,8,9], visceral hypersensitivity [10,11], dysmotility [12,13], intestinal mucosal immune system activation [14,15,16], inflammation, dysfunction of the gut–brain axis, alteration in the gut microbiome [17], food sensitivity, genetics, and psychological disturbance have been hypothesized as the possible mechanisms involved in IBS [18,19,20]. There is currently no specific treatment available for IBS, and most of the therapeutic management is supportive, aimed at relieving the individual symptoms. IBS routine care might include symptomatic drug therapies such as antispasmodics, prokinetics, laxatives, antidiarrhoeals, antibiotics, and antidepressants [21,22,23,24,25]. However, despite the availability of several treatments for IBS, they are often ineffective. Some classes of these drugs have not shown any clear efficacy in randomized clinical trials (RCTs) [26]. Besides pharmacological treatment, lifestyle changes including diet modification, such as intake of low-FODMAP (fermentable oligosaccharides, disaccharides, monosaccharides, and polyols) diet, fibre, probiotics supplementation, regular exercise, increased physical activity, adequate sleep, and psychotherapy have also been suggested to help improve the symptoms of IBS; however, evidence supporting these approaches is limited [27].

Due to the heterogeneity of IBS, it is highly unlikely that a single “magic bullet” drug will completely cure IBS. Instead, a combination of pharmacological agents could effectively manage the symptoms of IBS. In the presence of a limited therapeutic arsenal, there is a need to explore for safe, effective, affordable, and widely available treatment for IBS. As modern pharmacological studies continue to provide compelling evidence regarding the therapeutic efficacy and safety of potential agents, there is an increasing scientific and clinical interest in exploring botanicals as a possible pharmacological therapy for IBS [28]. According to the reported literature, supplementation of extracts from various botanicals, including *Mentha piperita* (peppermint), Aloe vera, *Curcuma longa* (turmeric) (Figure 1), *Cynara scolymus* (artichoke), *Fumaria officinalis* (common fumitory), *Berberis aristata* (Indian barberry) (Figure 1), *Hypericum perforatum* (St. John’s wort), *Plantago psyllium* (psyllium husks), *Melissa officinalis* (lemon balm), and *Ferula assafoetida* (assafoetida powder) have been studied for their potential therapeutic effect in the management of IBS symptoms [29,30]. Among these botanicals, several animal-model studies and some clinical trials have suggested that extracts from *Berberis aristata* (berberine or BBR) [31,32,33,34,35,36,37,38,39,40,41] and *Curcuma longa* (curcumin or CUR) [42,43,44,45,46,47,48] may be more efficacious in the management of IBS. Both BBR and CUR standardized extracts-based nutritional supplements are widely available and could provide a suitable supportive pharmacological option in the management of IBS symptoms.

Berberine (Figure 1) is an isoquinoline-based alkaloid found in the barks, leaves, twigs, rhizomes, roots, and/or stems of various plants, particularly in the Berberis genus, which includes plants like barberry (e.g., *Berberis aristata*), goldenseal, Oregon grape, greater celandine (*Chelidonium majus*), and Chinese goldthread [49]. Berberine extracts have been utilised for centuries in traditional Chinese and Ayurvedic medicine to treat a variety of health conditions, including GI infections, diarrhoea, and inflammation [50]. Berberine possesses diverse pharmacological properties, including but not limited to antimicrobial, anti-inflammatory, analgesic, antinociceptive, and antidepressant effects [31]. Berberine is suggested to alleviate symptoms of IBS by mitigating stress-induced intestinal inflammation and reducing visceral hypersensitivity [40]. Additionally, it moderates bowel motility while also regulating the composition of the gut microbiome [41].

Curcumin (Figure 1), a natural polyphenol, is the main curcuminoid found in turmeric, a popular spice used in food preparation in South Asia and the Middle East. Curcumin has long been used as a traditional medicine to treat a wide array of health conditions, including GI conditions, inflammatory diseases, stress, and mood disorders [51]. Modern pharmacological studies highlight CUR’s versatility, portraying it as a potent anti-inflammatory [52], antidepressant [53], and antioxidant agent [54]. Its pharmacological effects extend to inflammatory bowel diseases and functional GI disorders [55,56,57]. Curcumin has been suggested as a promising adjunct therapy for GI conditions owing to its broad impact on the GI system, including modulation of intestinal microbiota, enhancement of intestinal barrier function, and attenuation of gut inflammation and oxidative stress, as well as its efficacy against bacterial, parasitic, and fungal infections [58].

The potential synergistic pharmacological effect (combined effect) of BBR and CUR (hereafter referred as BBR/CUR) holds significant promise and has demonstrated efficacy across a spectrum of conditions. These encompass bacterial infections and associated inflammation [59,60], as well as the regulation of gut microbiota in obesity and the related gastrointestinal and hepatic alterations [61]. Furthermore, its potential benefits extend to cancer [62,63,64], non-alcoholic fatty liver disease (NAFLD) [65,66], and Alzheimer’s disease (AD) [67]. There is growing body of pharmacological evidence from in vivo studies that support the possible therapeutic effect of BBR and CUR in IBS. However, human clinical studies are needed to investigate the safety and efficacy of BBR and CUR in the management of symptoms associated with IBS. Given that multiple pathophysiological mechanisms are involved in IBS, we propose that the potential synergistic pharmacological effect associated with co-administration of BBR/CUR can provide an effective natural therapy for the management of IBS symptoms. In the present observational study, conducted under real-life routine clinical practice settings, for the first time, we aimed to assess this synergistic therapeutic effect of a dual-component supplement of standardized extracts of BBR and CUR (Enterofytol^®^ PLUS) as a complementary therapy in the management of IBS patients in the Belgian population. The results support the potential enhanced pharmacological effect of BBR/CUR supplementation, resulting in significant improvement in the patients’ IBS severity index (IBSSI) and QoL.

## 2. Materials and Methods

### 2.1. Study Design and Patient Population

This was a post-observational, retrospective, non-controlled, and non-randomized clinical study conducted in real-life routine clinical practice settings. The study involved the analysis of treatment history/clinical outcome of a cohort of 146 IBS patients (see consort flow diagram in Appendix A), enrolled and followed-up at Belgian general practitioners’ (GPs) clinics and pharmacies between 25 August 2020 and 16 May 2022. These patients were prescribed/advised of the BBR/CUR complementary therapy, as deemed appropriate by the clinician for the management of IBS symptoms, and this was not guided by any predetermined criterion. A total of 38 GPs and 3 pharmacies contributed to the data of these patients. The GPs and pharmacists enrolled patients sequentially as they arrived. There was no selection on any ground. Patients underwent two clinical assessments for IBS, occurring at the baseline visit (T_0_) when they received a prescription for standard of care IBS medication alone or were advised to take the BBR/CUR supplement as a complementary therapy (as two oral tablets daily for 2 months), and at the 2-month follow-up visit (T_1_). The patients’ demographic and clinical results were recorded by the physician or pharmacist on a “case report form” (CRF). All the patients had to purchase their own supplement. Furthermore, the dietary supplement was consumed in adherence with the good clinical practice (GCP), ensuring it did not alter the patients’ therapeutic regimens. Importantly, the decision to use the supplement, readily available over the counter in Belgium, was independent of the study and did not necessitate supplementary diagnostic or follow-up procedures. According to the Belgian law governing human subject experimentation, institutional review board (IRB) ethics approval was not required given the retrospective nature of the post hoc observational study, which involved the collection of the patients’ treatment history data from the GPs’/pharmacies’ files [68]. The study was registered in ClinicalTrials.gov, registration ID NCT06187298.

### 2.2. Patients’ Inclusion Criteria

Patients included in the study analysis had to meet the following criteria: (1) IBS diagnosis as per the Rome IV criteria; (2) had IBS symptoms that appeared before the age of 50 (to allow the exclusion of suspected cases of colorectal cancer); (3) had to be free of involuntary weight loss, family history of chronic inflammatory bowel disease, colorectal cancer, celiac disease, rectal discharge, nocturnal symptoms, fever, and abnormalities on clinical examination (abdominal mass, signs of anaemia); and (4) had used Enterofytol^®^ PLUS supplement (2 tablets a day for 2 months) as complementary therapy for IBS.

### 2.3. Variables

The demographic data of the patients, encompassing age, gender, and the source of information (either from a physician or pharmacist) along with the dates of their visits were meticulously recorded. The assessment of the patients’ IBS symptoms at both visits was structured around several parameters: (a) abdominal pain, quantified using a visual analogue scale (VAS) ranging from 0 (no pain) to 100 (very intense pain); (b) abdominal distension or tightness (bloating), measured on a VAS scale from 0 (no distension) to 100 (very intense distension); (c) satisfaction with intestinal transit, assessed using a VAS scale ranging from 0 (very satisfied) to 100 (very dissatisfied); and (d) evaluation of life interference due to IBS symptoms, gauged on a VAS scale from 0 (not at all) to 100 (completely). Other IBS symptoms recorded, include, (a) the minimum and maximum stool frequency (number per day/week/month); (b) the frequency (often/sometimes/never) of normal stools: hard, very fine, in small pieces, soft, and liquid; and (c) the occasional presence (yes/no) of mucus in stools, blood in stools, having faecal urgency, difficulty defecating, or a feeling of incomplete bowel emptiness. Information about IBS symptomatic treatments in progress or already tried included the following: (a) the type of treatment; (b) status: already tried and stopped or ongoing; and (c) dosage and frequency. Other IBS treatments were also recorded: (a) antidepressants/benzodiazepines (if yes, which ones and what dosage); (b) probiotics (if yes, which ones and what dosage); (c) specific dietary restrictions (if yes, which ones); and (d) other complementary treatment. Additionally, at the 2-month follow-up visit, the following were recorded: (a) compliance with the BBR/CUR supplement regime; (b) side effects: if yes, which ones; (c) treatment impact on QoL; (d) patient’s general satisfaction; (e) patient’s desire to continue supplement treatment (yes/no), (f) improvement and/or resolution of IBS symptoms (yes/no); and (g) doctor’s/pharmacist’s comments.

At both visits, the patient’s stool status was assessed according to the Bristol stool chart (BSC) [69]. The BSC is a medical tool utilised to classify the shape and consistency of human faeces. The chart classifies the stools into seven different types, ranging from hard lumps (Type 1) to entirely liquid (Type 7). According to BSC, Types 1 and 2 stools (hard and lumpy stool) may indicate constipation or slow transit in the digestive system; Types 3 and 4 stools suggest a healthy or normal bowel movement, while Types 5 (soft pieces/blobs), 6 (loose or mushy stool), and 7 (watery and liquid) may indicate a faster transit time in the digestive tract, potentially related to certain dietary factors or infections or potential issues with absorption in the intestines; Types 5, 6, and 7 tend towards diarrhoea. The BSC chart is widely utilised by healthcare providers for assessing bowel patterns and habits and diagnosing conditions such as constipation, diarrhoea, and IBS.

### 2.4. Study Outcome Measures

The main objective of the study was to assess the BBR/CUR supplement efficacy based on changes in the patients’ IBSSI, which was considered the primary outcome measure. The IBSSI is a reliable and well-validated instrument for measuring the presence and severity of IBS symptoms in clinical settings [70]. IBS severity index is a composite score of typical IBS symptoms, including (1) the severity of abdominal pain, (2) the severity of abdominal distention/tightness (bloating), (3) satisfaction with bowel habits, and (4) life interference due to IBS symptoms. Each measure is rated on a VAS from 0 to 100, with the overall score (IBSSI) ranging from 0 to 400. IBS severity is graded as mild (≤175), moderate (175–300), or severe (≥300) based on the overall score [70]. A 15% change in the IBSSI is considered clinically significant in measuring efficacy. The study’s secondary outcomes concerned the qualitative aspects of bowel movements, effect on stool, need for concomitant symptomatic conventional IBS treatments, supplement safety and tolerability, effect on QoL, and patients satisfaction.

### 2.5. Berberine–Curcumin (BBR–CUR) Supplementation

Each patient took the oral BBR/CUR supplementation at a dose of two tablets a day (one tablet after breakfast and one tablet after dinner) for 2 months. The BBR/CUR extracts-based supplement used in the study (Enterofytol^®^ PLUS, manufacturer Tilman SA, Baillonville, Belgium) is a nutritional supplement approved by the competent Belgian authorities (NUT/PL31/153; Service public fédéral, Santé, Sécurité de la chaîne alimentaire et Environnement). Each Enterofytol^®^ PLUS tablet contains 200 mg of standardized BBR extract from the roots of *Berberis aristata* and 49 mg of standardized CUR extract from *Curcuma longa* rhizome (CURTIL02) (equivalent to 42 mg of CUR) (Figure 1). The patients had the option of maintaining their conventional antispasmodic or stool-regulating (antidiarrhoeal or laxative) treatments for immediate relief of IBS as long as needed.

### 2.6. Statistical Analysis

As a retrospective study, no formal sample size calculation was made but investigators expected to include at least *n* = 100 patients in the study. A power calculation shows that with such a sample size and assuming a baseline IBSSI of 300 and a standard deviation (SD) of 80 points, a drop of 15% in IBSSI after treatment could be detected with at least 99% power at the 5% critical level. The qualitative variables were described using frequency tables, while the quantitative variables were summarized in terms of mean and standard deviation (SD) or median and interquartile range (IQR) for non-normal distributions. The data collected before and after supplementation were compared by Student’s paired *t*-test. The mean absolute differences were presented with their 95% confidence interval (95% CI) and *p*-values; relative differences (%) between pre- and post-treatment means were also given. Paired proportions were compared by the symmetry test and McNemar test, while the Chi-squared test and Fisher’s exact test were used to compare group proportions. Multiple linear regression was used to assess the influence of the covariates on the mean absolute differences. Ordinal logistic regression was used to assess the relationship between patient satisfaction and patient characteristics factors, with the results being expressed in terms of odds ratios (ORs) with the associated 95% CI and *p*-value. When the sample sizes were too small or in the case of estimation computational difficulties, a Haldane correction and Firth logistic regression were applied to the data to obtain reliable estimates. All the analyses were performed on the maximum available data, and missing values were neither replaced nor imputed. The results were considered significant at the 5% critical level (*p* < 0.05). Calculations and graphs were performed using Statistical Analysis System (SAS) software version 9.4 and R (version 4.3.1).

## 3. Results

### 3.1. Primary Outcome Measure

The patients’ mean age was 52 ± 19 years, and they were predominantly female (69%). The median (IQR) time between the two visits was 8.9 (8–10) weeks. The majority (91.1%) of patients had complied with the supplement two-tablets-a-day dose regime. The average duration of concomitant conventional IBS treatment was 53.3 ± 15.5 days. After approximately 2 months of taking the BBR/CUR supplement, the efficacy of BBR/CUR supplementation on the patients’ IBS severity was assessed by comparing the IBSSI before and after 2 months of supplementation. Figure 2 illustrates the overall distribution of the 146 patients according to the degree of their IBS severity level. The IBSSI was 255 ± 55 (IBS moderate degree symptoms) at T_0_ (Figure 2A) and it decreased to 134 ± 71 (IBS mild degree symptoms) at T_1_ (Figure 2B). The improvement between the two visits was highly significant (*p* < 0.0001, Table 1). The efficacy of the treatment based on the reduction of the IBSSI was not influenced by age or gender, but high IBSSI values at baseline led to a greater reduction of the severity index after treatment.

In terms of the supplement effect on individual symptoms, the score for all the four primary IBS symptoms decreased significantly (*p* < 0.0001) between the two visits, i.e., abdominal discomfort by 47.2%, abdominal distension by 48.0%, intestinal transit by 46.8%, and IBS influence on QoL by 48.1% (at least 84.9% of patients experienced a decrease in each symptom score) (Table 1). The same was also true for the patients’ frequency of stool passage per week (*p* < 0.0001). The cohort average number of weekly stool passage ranged from 7.3 ± 6.6 to 16.5 ± 11.9 at T_0_, and from 6.7 ± 4.3 to 12.0 ± 7.5 at T_1_ (Table 1). The weekly maximum stool passage frequency significantly decreased between the two visits (*p* < 0.0001).

### 3.2. Secondary Outcome Measures

#### 3.2.1. Supplementation Effect on Patients’ Stool Type (Status)

Figure 3 illustrates patients’ stool status at baseline and after 2 months of BBR/CUR supplementation. The patients’ stool description (according to BSC) also showed an improvement between T_0_ and T_1_. The improvement was statistically significant for all stool types: *p* < 0.0001 for normal (Figure 3A), *p* = 0.0005 for hard stools (Figure 3B), *p* = 0.0001 for fine stools (Figure 3C), and *p* < 0.0001 for pieces of stool (Figure 3D), loose stools (Figure 3E), and liquid stools (Figure 3F). At T_0_, the majority (58.2%) of patients reported only sometimes having normal stools. At T_1_, the majority (58.2%) patients reported that they often had normal stools, implying a significant improvement in stool consistency.

#### 3.2.2. Supplementation Effect on Patients’ Stool Situation

The patients’ stool situations also showed significant improvements between T_0_ and T_1_ (Figure 4). At T_0_, 32.6% of patients reported finding mucus in their stools (Figure 4A), and 9.4% found blood (Figure 4B). In addition, most patients had to rush to the toilet (58.0%) (Figure 4C), tried to have a bowel movement (53.6%) (Figure 4D), or felt that they had not completely emptied their bowels after a bowel movement (75.4%) (Figure 4E). At T_1_, the proportions of patients who found mucus or blood in their stools significantly decreased to 11.8% (*p* < 0.0001) (Figure 4A) and 2.2% (*p* = 0.0039) (Figure 4B), respectively. Moreover, most of the patients did not need to rush (71.0%, *p* < 0.0001) (Figure 4C), make an effort (66.7%, *p* < 0.0001) (Figure 4D), or have the impression of a non-empty bowel (65.9%, *p* < 0.0001) (Figure 4E).

#### 3.2.3. Supplementation Effect on IBS Sub-Types

The data of the patients’ IBS sub-types were not available and hence were not included in the study. However, medications, particularly those for “diarrhoea” (IBS-D) and “constipation” (IBS-C), were collected for each patient at both T_0_ and T_1_. This offered an opportunity to assess any variation in the impact of supplementary therapy on different IBS subtypes. Amongst the total 146 patients, 37 patients were treated with concomitant antidiarrhoeic medication and/or laxative medications. Of these, 8 had antidiarrhoeic drugs only (IBS-D patients), 17 had laxative drugs only (IBS-C patients), and 8 received both (Appendix A). The analysis of the primary outcome for the 8 IBS-D patients and the 17 IBS-C patients is given in Appendix A, respectively. In both cases (despite the small sample sizes), there was a highly significant decrease in primary IBS symptoms and overall IBSSI. The maximal weekly stool passage frequency decreased significantly for IBS-D patients (*p* = 0.0012), and the minimal weekly stool passage frequency dropped significantly for IBS-C patients (*p* = 0.0097). Thus, despite the approximate approach used here, the results mimic those obtained for the entire population.

#### 3.2.4. Supplementation Effect on Concomitant Conventional IBS Treatment

At baseline, 34.2% of patients were taking at least one antispasmodic treatment, 9.6% at least one treatment for diarrhoea, and 15.8% at least one treatment for constipation (Table 2). At the 2-month follow-up visit, the proportions of patients with at least one treatment decreased to 12.3% for antispasmodics, 3.4% for diarrhoea, and 13.7% for constipation (Table 2). In terms of the relative differences in means, this corresponds to a decrease of 64.0%, 64.6%, and 13.3% for antispasmodics, diarrhoea, or constipation, respectively (Table 2). As for the other treatments prescribed for IBS, such as antidepressants/benzodiazepines, probiotics, or dietary restrictions, the proportion of patients with antidepressants/benzodiazepines were 33.6% and 30.6% at T_0_ and T_1_, respectively, i.e., a relative mean reduction of 8.9% (Table 2). For probiotics, the patients’ proportions were 30.3% and 18.8% at T_0_ and T_1_, respectively, which corresponds to a relative mean decrease of 38.0% (Table 2). Finally, at T_0_, 38.9% of the participants had specific dietary restrictions, while at T_1_, this came to 27.8%, i.e., a relative mean reduction of 28.5% (Table 2).

The evolution of IBSSI was also compared between two groups of patients defined with respect to their concomitant conventional IBS treatment at T_0_ (Table 3). The results reveal no evidence of significant difference in any of the comparisons.

#### 3.2.5. Supplementation Safety and Tolerability

The BBR/CUR supplement complementary therapy was well tolerated by all the patients. A small number (*n* = 9 or 7.1%) of patients reported mild effects that included digestive, 5 (3.4%); mood, 1 (0.6%); itching, 1 (0.6%); other 1 (0.6%); and unspecified, 1 (0.6%). There were no serious adverse effects or treatment-emergent effects reported.

#### 3.2.6. Patients’ Satisfaction with Supplement Complementary Therapy

At baseline, patients were generally dissatisfied with their intestinal habits and believed that IBS had a negative impact on their life. Following 2 months of BBR/CUR supplement complementary therapy, 82% of the patients were satisfied with the management of their IBS symptoms. Most (93.1%) patients reported an improvement or even resolution of their abdominal discomfort, and 79% patients wanted to continue the supplementation. Patients who decided to continue the supplement complementary therapy were significantly more satisfied than the others (*p* < 0.0001) and corresponded to higher proportions of improvement and/or resolution of the initial problem (*p* < 0.0001). The patients’ satisfaction increased as the IBS severity decreased, both in terms of component symptoms and overall severity index (OR = 1.19, *p* < 0.0001). The increase in normal stools (OR = 5.23, *p* < 0.0001) (Figure 5A) and the decrease in fine stools (OR = 3.16, *p* = 0.0051) (Figure 5B), loose (soft) stools (OR = 2.40, *p* = 0.015) (Figure 5C), or liquid stools (OR = 2.70, *p* = 0.0064) (Figure 5D) had a positive impact on the patients’ satisfaction.

Furthermore, the disappearance of mucus in the patients’ stool (OR = 6.29, *p* = 0.014) (Figure 6A), the absence of urgency to rush to the toilet (OR = 7.32, *p* = 0.0003) (Figure 6B), overcoming difficulty in defecation (OR = 3.01, *p* = 0.023) (Figure 6C), or the absence of feeling of incomplete evacuation (OR = 5.92, *p* < 0.0001) (Figure 6D) were all the factors that positively influenced the patients’ satisfaction.

## 4. Discussion

This pragmatic clinical study assessed, for the first time, the potential synergistic pharmacological effect of a combined BBR/CUR standardized extracts supplementation in the management of a cohort of IBS patients (*n* = 146) under real-life routine clinical practice conditions. At baseline, the cohort average IBSSI was 255 ± 55, and after the 2 months of BBR/CUR supplementation, the IBSSI decreased to 134 ± 71, demonstrating a significant (*p* < 0.0001) clinical improvement/reduction (48%) in the overall IBS symptom severity of the patients. This overall IBSSI improvement was accompanied by a significant reduction (*p* < 0.0001) in patients’ all primary IBS severity parameters, including abdominal discomfort/pain, abdominal distension, and intestinal transit, as well as IBS influence on QoL. In addition, the supplementation also resulted in a decrease in the patients’ number of bowel movements without causing constipation. Moreover, the supplement complementary therapy also significantly improved (normalised) the patients’ stool status and situation, as well as leading to a 64% reduction in the need for some conventional IBS medications, such as antispasmodics and antidiarrhoeals. However, the patients’ reliance on antidepressant and laxative medications was not significantly influenced. Most of the patients (82%) were satisfied with the supplement complementary therapy, especially as they achieved relief in their IBS symptoms within a relatively short period of 8.9 weeks, without any noticeable side effects. Overall, these results support the potential synergistic pharmacological effect of BBR/CUR supplementation on the broad GI symptoms associated with IBS, at least within the time range (i.e., 2 months) observed in this study. The effect occurs irrespectively of age, sex, and IBS subtypes, although it was stronger when the initial severity of symptoms was higher. To our knowledge, this is the first clinical study that investigated the potential synergistic pharmacological effect of BBR/CUR supplementation in IBS patients, and reveals an improvement in the patients’ IBS severity index. This study also implies the excellent safety and tolerability of BBR/CUR supplement complementary therapy in the primary management of IBS.

The observed improvement in IBS symptomatology and QoL associated with the possible pharmacological effects of BBR/CUR are in line with results from previously reported clinical trials. While CUR has been investigated in several clinical trials [42,43,44,45,46,47,48], only a few trials have assessed the beneficial effects of BBR in the management of IBS [33,39]. In a study by Chen et al., in patients with IBS-D, 400 mg BBR hydrochloride supplementation twice a day for 2 months led to significant improvement in the patients’ diarrhoeal frequency (*p* = 0.032), abdominal pain frequency (*p* < 0.01), and defecation urgency frequency (*p* < 0.01) as compared to the placebo group [33]. In addition, BBR supplementation also improved the scores of the patients’ IBS symptoms, depression, anxiety, and QoL. In another study by Wang et al., in patients with IBS, BBR plus probiotic supplementation was associated with significant clinical efficacy through regulation of the inflammatory response [39].

The possible pharmacological effect of CUR, acting either alone or in combination with other botanicals, in the management of IBS has also been investigated in several clinical trials, and most of these studies have revealed beneficial effects of alleviation of IBS symptoms and improvement in QoL. In a study by Bundy et al., in patients with IBS, a daily intake of 72 mg or 144 mg CUR extract supplementation for 8 weeks significantly decreased the prevalence of IBS and abdominal discomfort/pain, as well as demonstrating improvement in the patients’ QoL [43]. There was also an improvement in other IBS symptoms and the patients’ bowel patterns [43]. In a study by Portincasa et al., a daily dose of a combined supplement of 84 mg CUR and 50 mg fennel essential oil for one month resulted in a significant decrease in the mean IBS symptoms severity index, as well as improvement in all IBS primary symptoms and QoL [47]. The proportion of symptom-free patients was significantly higher in the supplement group as compared to the placebo group. In another study on the same supplement combination, using a dose of 84 mg CUR and 50 mg fennel essential oil twice a day for one month, followed by the same dosage once a day for another month demonstrated a significant reduction in the patients’ IBS severity index, as well an improvement in their QoL [44]. In a study by Woźniak et al., a daily dose of 600 mg CUR supplementation for one and three months resulted in a significant improvement in the IBS symptoms severity index, especially for bloating and abdominal pain, as well as the patients’ QoL [48]. Other clinical trials have also revealed clinical benefits of CUR supplementation in IBS, including remission in patients’ digestive complaints, improvement of anxiety, and patient satisfaction with their bowel habits [45,46]. In contrast, in one study, a daily 180 mg *Curcuma xanthorriza* extract supplementation for 18 weeks did not show any therapeutic benefit over placebo in patients with IBS [42].

The possible beneficial effects of complementary therapy of BBR/CUR in alleviating symptoms of IBS is likely due to the synergy of multiple pharmacological mechanisms (combined effect) associated with BBR and CUR. According to the reported evidence, BBR can alleviate symptoms of IBS by multiple mechanisms [31], including (1) anti-inflammatory effects (via inhibition of the intestinal nuclear factor–kappa B (NF-kB) signalling pathway) and modulating pro-inflammatory cytokines, including tumour necrosis factor-alpha (TNF-α), interferon-gamma (IFN-γ), and interleukin-7 (IL-7) [71]; (2) regulating visceral hypersensitivity and intestinal motility [40,41,72] by reducing the expression of brain-derived neurotrophic factor (BDNF) and its receptors, tropomyosin receptor kinase B (TrkB) and C-kit; (3) enhancing intestinal mucosal barrier function [35,73,74]; (4) regulating composition of intestinal flora [36,41,73,74,75,76]; (5) inhibition of neurotransmission within colonic smooth muscle [37]; (6) improving intestinal epithelial tight junctions by upregulating A20 expression [34]; and (7) via its antinociceptive effect [38].

Curcumin has been suggested to alleviate symptoms of IBS through multiple anti-inflammatory mechanisms [52,77,78], including suppressing circulating IL-6 [79,80], NF-kB, and TNF-α [81], as well as regulating key mediators of cellular inflammation, including 5-lipoxygenase (5-LOX), cyclooxygenase-2 (COX-2), and inducible nitric oxide synthase (iNOS) [82]. Other pharmacological mechanisms suggested for CUR in IBS include regulating the brain–gut axis (by increasing serotonin (5-hydroxytryptamine or 5-HT), BDNF, and phosphorylation of cyclic adenosine monophosphate (cAMP) response element-binding protein (pCREB) expression in the hippocampus and colon [83,84]) and restoring intestinal barrier integrity (increased expression of tight junction proteins zonula occludens-1 (ZO-1) and occluding), as well as altering the overall composition of the gut microbiota [85].

Both BBR and CUR have been investigated in diverse pharmacological studies for their potential beneficial effects in managing IBS symptoms. Below are some of the strengths and limitations of BBR and CUR as compared to other botanicals or conventional IBS medications.


*Berberine*


Anti-diarrhoeal effects: BBR has been shown to possess anti-diarrhoeal properties, which may be beneficial for IBS patients experiencing diarrhoea-dominated symptoms.Antimicrobial and antispasmodic effects: BBR exhibits broad-spectrum antimicrobial activity, including against pathogenic bacteria and parasites, which may help rebalance gut microbiota in IBS patients. It also exhibits antispasmodic effects, potentially reducing IBS-related abdominal pain.Regulation of gut motility: BBR has been reported to modulate gut motility, potentially alleviating symptoms of abdominal pain and discomfort associated with IBS.Choleretic effect: BBR may increase bile production and secretion, which may aid in the digestion of fats, and thus could alleviate some digestive symptoms commonly associated with IBS, such as bloating, gas, and discomfort. Bile acids have been shown to influence gut motility, and abnormalities in bile acid metabolism have been implicated in IBS. By promoting bile secretion, BBR may help regulate gut motility, potentially reducing symptoms like diarrhoea or constipation in individuals with IBS.Metabolic benefits: BBR has shown benefits in managing blood sugar levels and lipid profiles, which may indirectly impact IBS symptoms [86].While some clinical studies have investigated the efficacy of BBR in IBS, the overall evidence base is still limited compared to conventional treatments. BBR may cause GI side effects, such as nausea, vomiting, and abdominal discomfort, in some individuals. BBR may interact with certain medications, including those metabolised by the liver or affecting heart rhythm.


*Curcumin*


Anti-inflammatory properties: CUR has potent anti-inflammatory effects, which may help alleviate gut inflammation associated with IBS symptoms.Antioxidant properties: CUR exhibits antioxidant properties, which can help neutralise free radicals and reduce oxidative stress, potentially contributing to the improvement of IBS symptoms.Gut health support: CUR may promote gut health by modulating gut microbiota and improving intestinal barrier function.Cellular communication: CUR influences cellular communication pathways, potentially mitigating intestinal bleeding, ulcers, and irritation within the digestive tract.Psychological benefits: Some studies suggest CUR may reduce anxiety associated with IBS [46].Choleretic effect: CUR exhibits choleretic properties, stimulating bile production and secretion. This can aid in the digestion of fats, potentially alleviating symptoms such as bloating and discomfort in individuals with IBS.Low toxicity: CUR is generally considered safe and well-tolerated, even at high doses.Limitations associated with CUR include its poor bioavailability, and limited clinical evidence. While promising, clinical evidence supporting CUR’s efficacy in IBS remains limited. More robust clinical trials are needed.

Compared to other botanicals or conventional IBS medications, BBR and CUR offer the advantage of being natural compounds with potentially fewer adverse effects. Curcumin and berberine can improve the symptoms of IBS through multiple simultaneous mechanisms in the gut. However, their efficacy and safety profiles need to be further elucidated through well-designed clinical trials. Additionally, their poor bioavailability and potential interactions with other medications should be taken into consideration when considering their use as part of IBS treatment. Overall, while direct clinical evidence may be limited, the combination of BBR and CUR holds promise for producing synergistic therapeutic effects in the management of IBS symptoms based on their complementary mechanisms of action, such as anti-inflammatory, anti-microbial, and gut-modulatory properties. This synergy enhances their ability to reduce gut inflammation, regulate gut microbiota, improve gut motility, and enhance intestinal barrier function, leading to more comprehensive relief of IBS symptoms, such as abdominal pain, bloating, and irregular bowel movements. Additionally, their combined action may provide a greater therapeutic effect compared to using either compound alone, potentially offering better outcomes for individuals with IBS.

The present study also implies the excellent safety and tolerability of BBR/CUR complementary therapy in the management of IBS and is consistent with the above-reported clinical trial studies. Berberine is generally well tolerated with a good safety profile [87]. The dosage of BBR used in various studies has been between 0.3 and 3.0 g per day. However, the dosage that has been frequently used in clinical trials is 500 mg, three times a day. In some clinical trials, BBR has been used for up to two years [88]. Some people may experience minor GI side effects from BBR [89]. In a recent meta-analysis study of 44 RCTs involving 4606 patients with cardiovascular diseases, no serious adverse reaction was reported with BBR supplementation [90]. The safety and tolerability of CUR is well established in humans. As per the U.S. Food and Drug Administration (FDA) classification, turmeric is generally recognized as safe (GRAS) [91] for human use. In a Phase I clinical trial in Taiwan, CUR supplementation up to 8 g/day for 12 weeks has been reported to be well tolerated in patients with precancerous conditions or non-invasive cancer [92]. Another clinical trial in the UK found that CUR dosage from 0.45 to 3.6 g/day for 16 weeks was generally well tolerated by people with advanced colorectal cancer [93].

We recognize the inherent limitations of our study, which can be addressed in well-designed and adequately powered future studies. The observational, non-randomised, uncontrolled, open-label, short-term design and limited IBS patient population are some of the limitations of this study. The absence of a control group precludes the assessment of potential confounding effects, such as diet/lifestyle, concomitant conventional IBS treatment, duration of illness, etc., during the 2-month observational period. Conversely, the study’s strength lies in its execution in real-life GP clinics or pharmacies’ routine clinical practice settings, involving a diverse group of IBS patients with broad inclusion criteria, allowing the generalization of the impact, and it reflects the overall patient satisfaction and compliance. The significant decrease in the IBSSI was confirmed after adjusting for age, gender, and initial symptoms severity scores. To reduce information and interpretation bias, the patients were simply enrolled sequentially, and the objective of the BBR/CUR supplement complementary treatment was clearly explained to each patient at the inclusion visit. As for clinical relevance, a drop of 150 points in the IBSSI can be considered potentially clinically relevant. Additional research is required to evaluate the possibility of progressive clinical improvement following more extended treatment and/or a higher/lower dosage of the investigated nutraceutical. Further studies should also aim to explore more comprehensively the therapeutic impact of the proposed nutraceutical combination on sub-types of IBS. As the current study did not include post-2-month treatment withdrawal follow-up, future research should investigate symptom recurrence after therapy suspension, and considering previous findings of beneficial effects from BBR and CUR when administered separately in IBS patients, should evaluate whether combining these components yields superior outcomes compared to their individual use.

## 5. Conclusions

In conclusion, the present pragmatic clinical study conducted in real-life GP clinical practice/pharmacy settings supports the potential synergistic pharmacological effect of BBR/CUR, and its co-supplementation may help in the management of IBS symptoms at least within the time range (i.e., 2 months) observed in this study. The effect occurs irrespectively of age, sex, and IBS sub-types, and it is even stronger for initially severe symptoms. The study also validates the excellent safety and tolerability of BBR/CUR complementary therapy in the management of IBS. Further clinical and experimental research should focus on further elucidating the underlying mechanism of this nutraceutical combination.

## Figures and Tables

**Figure 1 nutrients-16-01204-f001:**
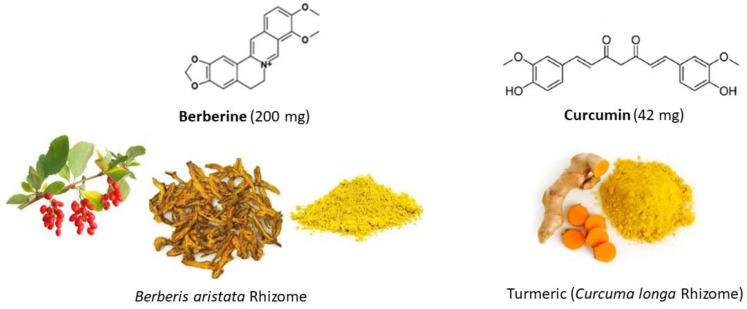
Chemical structure of berberine (BBR) and curcumin (CUR), the botanical bioactive agents, assessed in the present study (Enterofytol^®^ PLUS) as an oral synergistic complementary therapy in the management of gastrointestinal symptoms of irritable bowel syndrome patients.

**Figure 2 nutrients-16-01204-f002:**
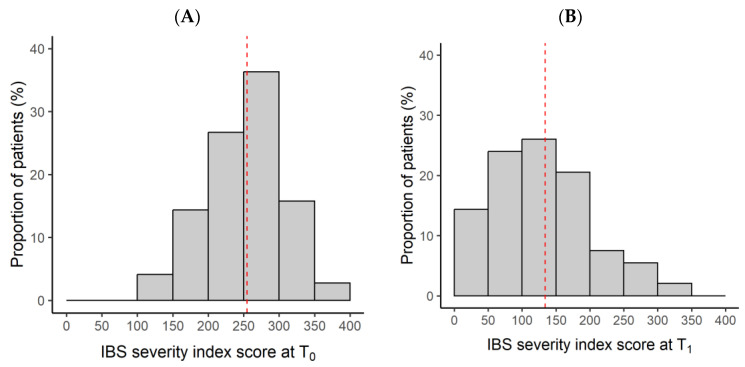
Distribution of patients’ IBS severity index (IBSSI) at baseline visit (**A**) and after 2-month BBR/CUR supplementation visit (**B**) (*n* = 146). The vertical dotted line (red) denotes the cohort average IBS severity index.

**Figure 3 nutrients-16-01204-f003:**
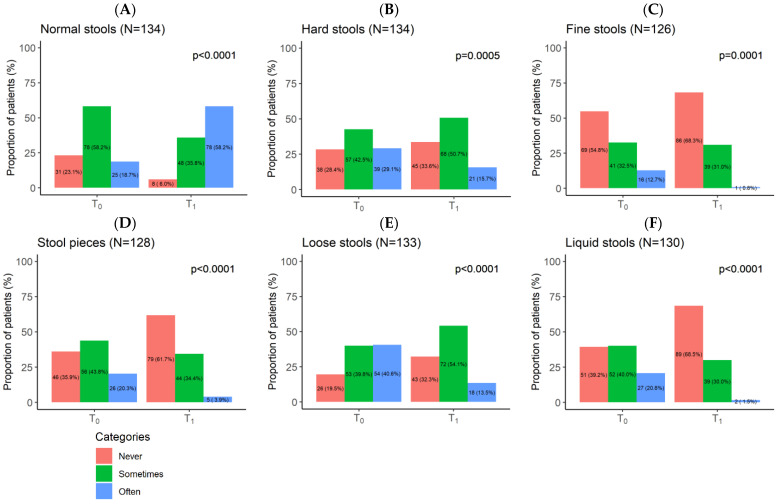
Berberine/curcumin (BBR–CUR) supplementation effect on IBS patients’ stool status: normal (**A**), hard (**B**), fine (**C**), pieces (**D**), loose (**E**), and liquid stools (**F**). T_0_: baseline visit, T_1_: after the 2-month BBR–CUR supplementation follow-up visit.

**Figure 4 nutrients-16-01204-f004:**
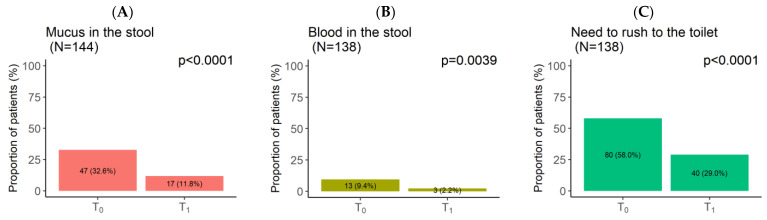
Berberine–curcumin (BBR–CUR) supplementation effect on patients’ stool situation at T_0_ and T_1_: mucus in the stool (**A**), blood in the stool (**B**), need to rush to the toilet (**C**), have to make an effort (**D**), and feeling of no emptiness of bowel (**E**). T_0_: baseline visit, T_1_: after the 2-month BBR–CUR supplementation follow-up visit.

**Figure 5 nutrients-16-01204-f005:**
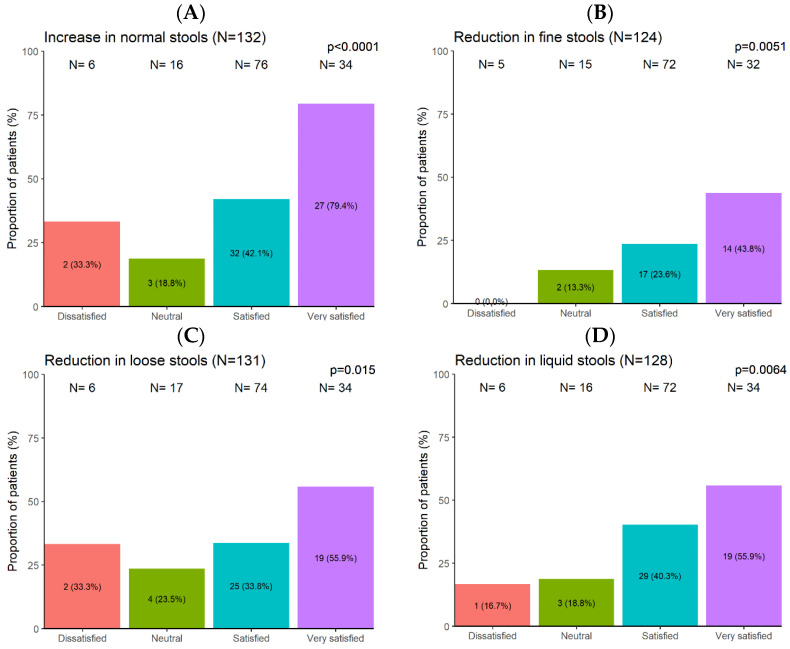
Association between patients’ supplement treatment satisfaction and patients’ stool type: increase in normal stools (**A**), reduction in fine stools (**B**), reduction in loose stools (**C**), reduction in liquid stools (**D**).

**Figure 6 nutrients-16-01204-f006:**
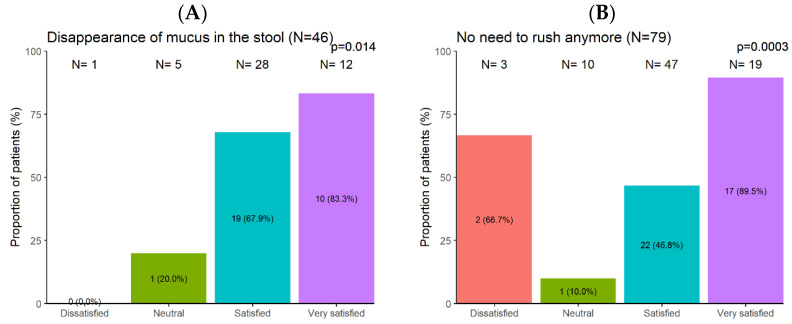
Association between patients’ treatment satisfaction and stool status and situation: disappearance of mucus in the stool (**A**), no need to rush anymore (**B**), no more straining to have a stool (**C**), disappearance of the non-empty feeling (**D**).

**Table 1 nutrients-16-01204-t001:** Patients’ IBS symptoms severity (mean ± SD) at T_0_ and T_1_ (*n* = 146).

IBS Symptoms	*n*	Score at T_0_	Score at T_1_	Difference T_1_−T_0_ (95% CI)	*p*-Value	Change (%)
Abdominal pain	146	60.2 ± 19.8	31.8 ± 19.6	28.4 (25.1–31.7)	<0.0001	47.2
Abdominal distension	146	64.0 ± 20.1	33.3 ± 20.8	30.7 (26.9–34.4)	<0.0001	48.0
Intestinal transit	146	66.0 ± 19.1	35.1 ± 20.8	30.9 (27.1–34.8)	<0.0001	46.8
Influence on quality of life	146	65.1 ± 17.2	33.8 ± 20.2	31.3 (27.8–34.8)	<0.0001	48.1
IBSSI	146	255 ± 55	134 ± 71.1	121 (109–133)	<0.0001	47.5
Weekly stool passage frequency						
Minimum	128	7.3 ± 6.6	6.7 ± 4.3	0.58 (−0.43–1.58)	0.26	8.2
Maximum	129	16.5 ± 11.9	12.0 ± 7.5	4.57 (2.94–6.21)	<0.0001	27.3

T_0_: baseline visit, T_1_: after 2-month BBR–CUR supplementation follow-up visit.

**Table 2 nutrients-16-01204-t002:** Supplementation effect on patients’ concomitant conventional IBS treatment at T_0_ and T_1_ (*n* = 146).

Concomitant IBS Treatment	Number of Medications or Use	T_0_	T_1_	Percent Decrease in Patients on Concomitant Treatment
		*n* (%)	*n* (%)	
Antispasmodic medication				64.0
	0	96 (65.8)	128 (87.7)	
	1	43 (29.4)	14 (9.6)	
	2	7 (4.8)	3 (2.0)	
	3		1 (0.7)	
Antidiarrhoeic medication				64.6
	0	132 (90.4)	141 (96.6)	
	1	13 (8.9)	3 (2.0)	
	2	1 (0.7)	2 (1.4)	
Laxative medication				13.3
	0	123 (84.2)	126 (86.3)	
	1	19 (13.1)	19 (13.0)	
	2	4 (2.7)	1 (0.7)	
Antidepressants benzodiazepines				8.9
	No	97 (66.4)	100 (69.4)	
	Yes	49 (33.6)	44 (30.6)	
Probiotics				38.0
	No	101 (69.7)	117 (81.2)	
	Yes	44 (30.3)	27 (18.8)	
Food restriction				28.5
	No	88 (61.1)	104 (72.2)	
	Yes	56 (38.9)	40 (27.8)	

T_0_: baseline visit, T_1_: after the 2-month BBR–CUR supplementation follow-up visit.

**Table 3 nutrients-16-01204-t003:** Comparison of IBSSI evolution (mean ± SD decrease between T_0_ and T_1_) with respect to concomitant conventional IBS treatment.

Concomitant Treatment	Without Treatment		With Treatment		
	*n*	IBSSI Score	*n*	IBSSI Score	*p*-Value
Antispasmodic	96	119.5 ± 73.5	50	124.6 ± 74.1	0.70
Antidiarrhoeic medication	132	119.1 ± 75.6	14	141.7 ± 46.9	0.27
Laxative medication	123	125.0 ± 74.8	23	101.3 ± 64.2	0.16
Antidepressants/benzodiazepines	97	124.3 ± 71.3	49	115.2 ± 78.0	0.48
Probiotics	101	118.8 ± 75.9	44	126.0 ± 69.0	0.59

T_0_: baseline visit, T_1_: after the 2-month BBR–CUR supplementation follow-up visit.

## Data Availability

The data supporting the findings of this study are included in the manuscript; further queries can be directed to the corresponding author.

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
