# Peer review of "The Possible Synergistic Pharmacological Effect of an Oral Berberine (BBR) and Curcumin (CUR) Complementary Therapy Alleviates Symptoms of Irritable Bowel Syndrome (IBS): Results from a Real-Life, Routine Clinical Practice Settings-Based Study"

_nutrients, 2024, doi:10.3390/nu16081204_

Round 1

Reviewer 1 Report

Comments and Suggestions for Authors

Current study evaluates the therapeutic effect of berberine (BBR) and curcumin (CUR) (Enterofytol® PLUS) supplementation as complementary therapy in the management of IBS patients in Belgian population. IBS is a debilitating condition, and it is becoming a major health concern. Effective therapies are essential to manage this condition. Thus, this study is relevant and addresses an important issue. However, this study requires some minor corrections.

1.      English should be checked by a native speaker and should be corrected.

2.      Error bars should be included in figures.

3.      Some background about the therapeutic use of berberine and curcumin and the biological basis of that use should be included in the introduction also. Overall, introduction should be formatted better for easy understanding of readers.

Comments on the Quality of English Language

English needs to be corrected in the whole manuscript.

Author Response

Dear Reviewer,

Many thanks. Please find attached our responses.

Kind regards

Reviewer 2 Report

Comments and Suggestions for Authors

1.      Each Enterofytol® PLUS tablet contains 200 mg of standardized BBR extract from the roots of Berberis aristata and 49 mg of standardized CUR extract from Curcuma longa rhizome (CURTIL02) (equivalent to 42 mg of CUR). 200 mg of standardized BBR extract was equivalent to how many mg of BBR?

2.      Enterofytol® PLUS” is a drug in market or in clinic trial?

Author Response

(The authors gave the same response as above.)

Reviewer 3 Report

Comments and Suggestions for Authors

The study's exploration of the efficacy of BBR/CUR supplementation in the symptom management of IBS is intriguing, particularly as data on the effectiveness of Traditional Chinese Medicine, including herbal treatments, in IBS therapy accumulates yet remains insufficient. My comments on this paper are as follows:

1. The study does not mention data on physical characteristics such as patients' body composition or weight. Given the variability in IBS symptoms and treatment responses among individuals, elucidating how these factors might influence the efficacy of BBR/CUR supplementation is crucial. Could this data be incorporated?

2. Within the data concerning the discontinuation of existing IBS medications, the usage of antidepressants/benzodiazepines does not appear to have decreased as much as other medications. Please discuss, within the discussion section, which specific symptoms BBR/CUR is particularly effective against in the treatment of IBS.

3. Regarding IBS-C patients, it is noted that "The maximal weekly stool passage frequency decreased significantly for IBS-D patients (p=0.0012) and the minimal weekly stool passage frequency dropped significantly for IBS-C patients (p=0.0097)." However, there is no mention of data for the weekly stool passage frequency at T1 in Table S3.

Author Response

(The authors gave the same response as above.)

Reviewer 4 Report

Comments and Suggestions for Authors

In my opinion, the submitted manuscript „ The possible synergistic pharmacological effect of an oral Berberine (BBR) and Curcumin (CUR) supplement complementary therapy alleviates symptoms of Irritable Bowel Syndrome (IBS): Results from a real-life routine practice conditions exploratory study” meets aims and scope of „Nutrients” Journal and Special Issue Natural Products and Human Health and may be accepted after the revision.

1.       The biggest objection of the presented research is the poorly designed study, which is supposed to resemble a clinical trial. Taking into account the fact that 2 authors of the publication are employed in Laboratoire Tilman, where Enterofytol plus is produced, the results of the presented studies, which also omitted the control group and the use of a double-blind trial, may be treated as of little scientific value. Moreover, in line 165, there is an information, that all patients had to purchase their own supplement – there is no information whether the purchase of the supplement by study participants was verified.

2.       There is no information that berberine and cucumin have also choleretic effect.

3.       Berberibe occurs also in Greater celandine (Chelidonium maius) – line 111.

4.       If you name curcumin as diferuloylmethane (line 120), you may write that it is a dimeric derivative of ferulic acid (curcumin is phenolic acid).

5.       When writing about T1 and T0, it might be better to write the numbers in subscript.

6.       There is no list of abbreviations in the article.

Author Response

(The authors gave the same response as above.)

Round 2

Reviewer 4 Report

Comments and Suggestions for Authors

The authors have made the suggested corrections to the manuscript of the publication and have also responded to the reviewer's comments, so I believe that the revised manuscript is suitable for publication.